# Multi-project wafer runs for electronic graphene devices in the European 2D-Experimental Pilot Line project

Bárbara Canto [1], Martin Otto [1], Arantxa Maestre[2,5], Alba Centeno[2], Amaia Zurutuza [2], Bianca Robertz[1], Eros Reato[3], Bartos Chmielak [1], Stefanie L. Stoll[1], Andreas Hemmetter [1,3], Florian Schlachter[1], Lisa Ehlert[1], Sha Li[1,6], Daniel Neumaier [1,4], Gordon Rinke [1], Zhenxing Wang [1] ✉ & Max C. Lemme [1,3] ✉

The commercialization of electronic devices based on graphene has not yet been successful, even 20 years after its first isolation. To this end, the European Commission is supporting research toward establishing a European experimental pilot line for electronic and optoelectronic devices based on graphene and related two-dimensional (2D) materials, namely the Experimental Pilot Line (2D-EPL) project. Here, we report the results obtained during the first and third multi-project wafer (MPW) runs completed at the end of 2022 (MPW run 1) and 2023 (MPW run 3) as an outcome of the 2D-EPL. Test devices were measured across the wafers to assess the device quality and variability before delivering the fabricated dies to the customers. Raman spectroscopy confirmed minimal structural changes in the graphene caused by the fabrication process, while electrical measurements of two different device types verified the device specifications defined in the process design kit.

Since graphene was first demonstrated to be a viable electronic material in 2004[1], considerable efforts have been made toward large-scale manufacturing technologies and reliable integrated devices[2]. The large-scale synthesis of graphene by chemical vapor deposition (CVD) on copper foils was first demonstrated as early as 2009[3,4], soon followed by large-scale graphene transfer from the growth substrate to target substrates and the fabrication of many different functional electronic devices[4–9]. Despite this, the semiconductor industry has not yet produced commercially viable graphene devices. In fact, several issues prevent moving from scientific experiments involving only a few graphene devices to wafer-scale processes with high yield and homogeneity, where the devices not only demonstrate a new functionality but also have characteristics within predefined specifications. In addition, there are challenges due to the incompatibility of graphene technology with certain standard processes in silicon technology. We believe the major difficulties are graphene growth[3,10], graphene transfer and cleaning[10–15], especially the removal of polymer residues from polymer-assisted transfer[10–12,15] and lithography processes[13,14,16,17], (dielectric) interfaces[18,19], doping[20–24], contact and ohmic contact formation[19–21,23,24], design[25], poor adhesion with metals[21–23,26], vulnerability to damage in plasma[22] and the choice of suitable dielectrics and their deposition[18,22,23,25,27].

The 2D Experimental Pilot Line (2D-EPL) project[28] addresses these engineering challenges with an ecosystem of tool providers and pilot line facilities in Europe, funded from October 2020 until September 2024 by the European Commission with a total of 20 million euros. The 2D-EPL emerged from the Graphene Flagship project with the purpose of both addressing these remaining challenges and establishing a first experimental pilot line for the prototype production of graphene and related two-dimensional (2D) material-based electronics, photonics,

[1]AMO GmbH, Aachen, Germany. [2]Graphenea S.A., San Sebastián, Spain. [3]RWTH Aachen University, Chair of Electronic Devices, Aachen, Germany. [4]University of Wuppertal, Chair of Smart Sensor Systems, Wuppertal, Germany. [5]Present address: ASM International, Leuven, Belgium. [6]Present address: Heraeus Precious Metals GmbH & Co. KG Herauesstr. 12-14, Hanau, Germany. ✉e-mail: wang@amo.de; lemme@amo.de

and sensors on a wafer scale[29]. In fact, one of the targets of the 2D-EPL is the development of industry-level process modules. There is also a second aim of the 2D-EPL that targets the delivery of graphene-based devices to customers. These two goals cannot be achieved simultaneously because their timelines do not match, i.e., production-ready tools and processes for wafer-scale transfer were not yet available for these MPW runs. Instead, the two goals were pursued in parallel. In this second target the availability of multi-project wafer runs (MPW runs) as a service was one of the most challenging goals, which were offered by different 2D-EPL partners with suitable clean room facilities. AMO was the lead institution in two completed MPW runs; the first run (2D-EPL MPW run 1) was completed in December 2022, and the second one, which was the third overall within the 2D-EPL project, was completed in October 2023 (2D-EPL MPW run 3). The availability of such MPW runs marks a significant step in increasing the maturity of wafer-scale graphene technology. MPW run 1 was intended mainly for graphene-based sensors, while MPW run 3 focused on graphene electronics.

Here, we report on the experiences and results of the two MPW runs. We first briefly describe the MPW run procedure before presenting and discussing the statistical data obtained from the material characterization and electrical device measurements. Finally, we will discuss some of the key challenges faced during fabrication, along with lessons learned for future MPW runs.

## Results
### MPW Procedure and design
The MPW runs were organized with an application phase (February to June 2022 for MPW run 1 and April to June 2023 for MPW run 3), during which specification sheets with expected performance parameters for graphene devices were made available to potential customers. In particular, graphene field effect transistors (GFETs) were defined both with areas of exposed graphene channels for chemical and biosensor applications and with dielectric encapsulation for electronic applications. The specified device performance parameters for mobility, charge neutrality point (CNP, also referred to as the Dirac voltage), sheet resistance ($R_{SH}$), and contact resistivity ($R_C. W$) are listed in Table 1. If the target values were reached and the optical microscopy analysis showed an acceptable quality regarding the lift-off and etching, we considered the wafer ready for delivery. Table 1 was defined in the initial 2D-EPL project description as specifications for potential customers and published on the 2D-EPL website[28]. Although higher values can and have been achieved, the specifications were set conservatively to ensure we could deliver what we promised to the customers. One parameter that was not included in the table is uniformity because the table was created for the benefit of the customers, who only ordered single chips from the MPW run, so uniformity was of secondary importance to them. Nevertheless, wafer scale uniformity is a very important parameter for industrial processes and was, therefore, measured and will be discussed in this section.

The potential customers first enrolled through a universal 2D-EPL access page[28] and then received a flexible process design kit (PDK) with information on the available materials for contacts and dielectrics, the respective fabrication methods, and the design rules for the lithography layers, e.g., critical dimensions. For MPW run 1, we deliberately chose to offer high design flexibility at the cost of a less automated PDK, which was appreciated by most customers. However, this large degree of design flexibility also raised process-related challenges, which will be discussed later; therefore, the PDK for MPW run 3 was more restrictive. After the application phase, the submitted chip designs were discussed individually with each customer. Then, all the customers' dies were combined into a common mask design, and the required lithography masks were ordered. The mask also contained test dies with reference devices spread evenly across the wafer, in addition to the customer dies, to monitor process quality on every wafer of the run.

The fabrication period was scheduled from September to November 2022 for MPW run 1 and from August to October 2023 for MPW run 3. During this period, more than 75 wafers were processed in total. These wafers were not finalized, instead we used them to optimize some of the fabrication steps. The dies were delivered to the customers between December 2022 and January 2023 for MPW run 1 and October 2023 for MPW run 3. Along with the chip delivery, each customer also received an individual report on the fabrication process, including Raman spectroscopy data, electrical measurements of the reference devices placed at various locations on the wafer, and optical micrographs of representative structures of their design and dies.

The two MPW runs received 60 applications from potential customers from nearly all continents, which included universities, start-ups, small and medium enterprises (SMEs), and large industries. Nineteen entities completed their orders, including all the target groups mentioned above. Their application areas and corresponding device designs include electronic devices, bio and terahertz (THz) sensors, and photodetectors. This aligns well with the intended goal of the 2D-EPL European Project, as these areas were explicitly mentioned in the initial proposal call text.

### Fabrication workflow
The design rules and the fabrication workflow of the PDKs for both runs targeted GFETs with local back gates and $Al_2O_3$ encapsulation (where applicable). The basic steps for fabricating the reference GFETs are briefly described here, with details available in the Methods section. Figure 1 depicts the workflow and a schematic cross-section of the reference devices for both MPW runs. The main process steps are the fabrication of the bottom gate (palladium (Pd)/titanium (Ti): 40 nm/5 nm), aluminum oxide ($Al_2O_3$) dielectric deposition, bottom contact deposition (Pd/Ti: 30 nm/5 nm) for MPW run 1 or nickel (Ni) adhesion pads (25 nm) for MPW run 3, graphene transfer, graphene patterning, and the deposition of top contacts (Pd 40 nm) and encapsulation ($Al_2O_3$) with optional channel opening for MPW run 1. In addition, vias to the metal contacts were fabricated through the gate dielectric and encapsulation.

MPW run 1 was carried out on p-doped 1–10 $\Omega$·cm Si wafers with 90 nm thermally grown $SiO_2$, while MPW run 3 used intrinsic, highly resistive (5 k$\Omega$·cm) Si wafers, also with 90 nm $SiO_2$. The $Al_2O_3$ dielectric

**Table 1 | Target and achieved values for MPW runs 1 and 3**

| Parameters | | Target values MPW run 1 and MPW run 3 | Achieved values MPW run 1 | Achieved values MPW run 3 |
|---|---|---|---|---|
| **Graphene Mobility** | | >1000 cm²/Vs | 1023 cm²/Vs | 1185 cm²/Vs |
| **Avg. Sheet Resistance** | $n_s = -10 \times 10^{12}$ cm$^{-2}$: | ~1 k$\Omega$ /sq | 0.72 k$\Omega$/sq | 0.6 k$\Omega$/sq |
| | at CNP: | ~4 k$\Omega$ /sq | 4.3 k$\Omega$/sq | 5.2 k$\Omega$/sq |
| **Avg. Contact Resistivity** | $n_s = -10 \times 10^{12}$ cm$^{-2}$: | ~1 k$\Omega$·µm | 0.8 k$\Omega$·µm | 1.5 k$\Omega$·µm |
| | at CNP: | ~4 k$\Omega$ µm | 3.5 k$\Omega$·µm | 5.3 k$\Omega$·µm |
| **Dirac Voltage** | | <15 V | − 2.85 V(forward)/ 37 V(backwards) | 6.5 V (forward)/ 7 V(backwards) |
| **Yield:** | | >80 % | 94 % | 87 % |

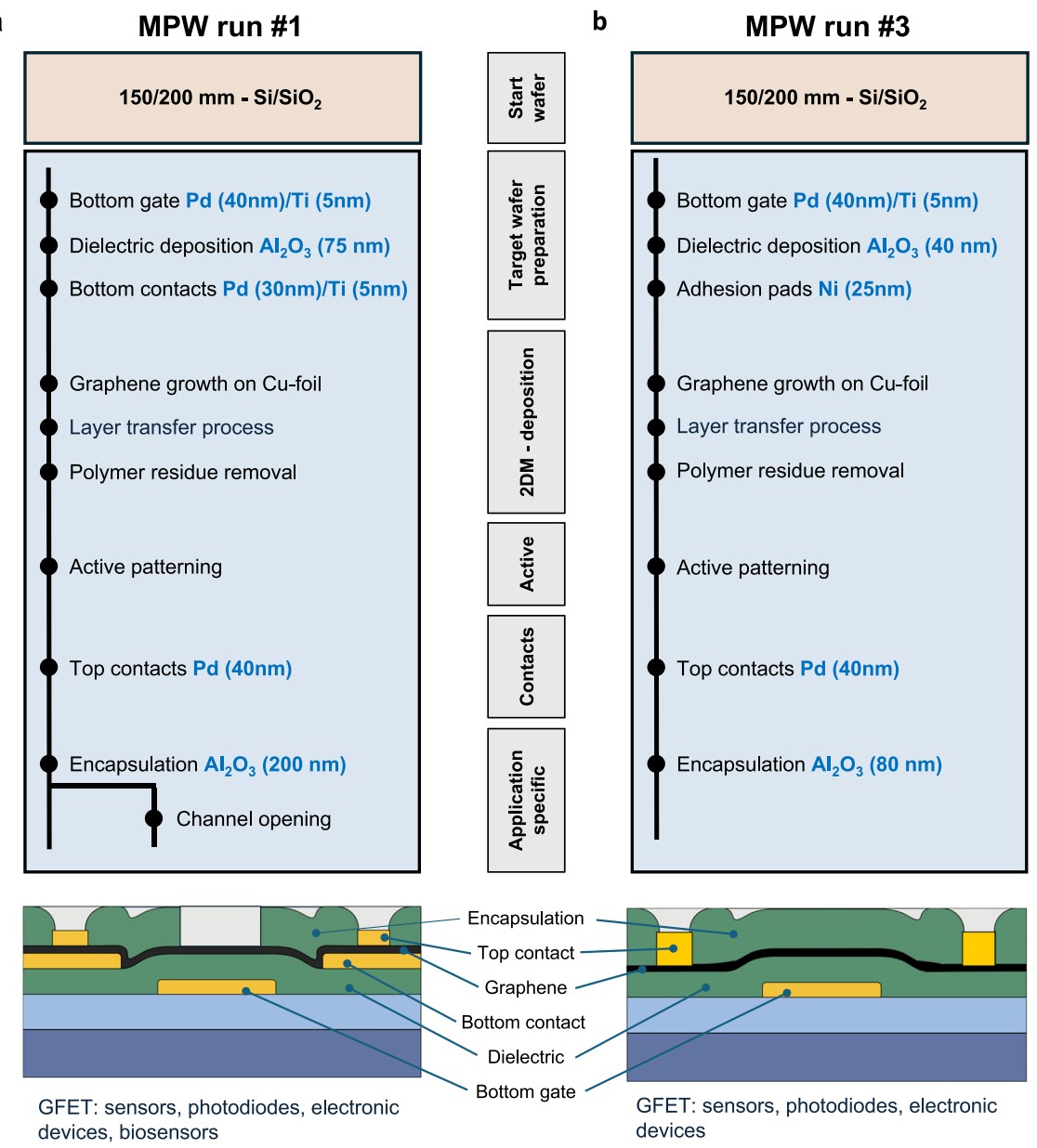

**Fig. 1 | Schematic diagram of the Multi Project Wafer (MPW) workflow for a graphene field effect transistor (GFET). a** Schematic diagram of the MPW run 1 workflow. **b** Schematic diagram of the MPW run 3 workflow. In both cases the 2D Material (2DM) used was CVD-graphene.

thickness, deposited by plasma-enhanced atomic layer deposition (PEALD), was 75 nm for MPW run 1 and 40 nm for run 3. The thickness decrease was implemented based on feedback from several customers of MPW run 1, who required lower operating voltages[30,31]. We also used different contact schemes using the same metal, Pd. In MPW run 1, the graphene was sandwiched between the bottom and top contacts[32], while in MPW run 3, only the top contacts were used. This was done to reduce complexity and accelerate processing, even though the contact resistance might be higher for top contacts than sandwich contacts. The final $Al_2O_3$ encapsulation step was also different in both runs. MPW run 1 offered the opening of the graphene area for sensing applications. This limited the process options because reactive ion etching (RIE) of the encapsulation on top of the graphene would have damaged the graphene. Wet chemical etching was unsuitable because the dielectric layer below the graphene was the same as the encapsulation layer and would have been attacked through potential cracks or holes in the graphene. We, therefore, chose a lift-off process for opening the graphene area and vias to the contacts, which required the use of

directional e-beam evaporation instead of conformal ALD growth. The encapsulation was 200 nm thick for MPW run 1. The encapsulation for MPW run 3 did not require openings above the graphene channel. Hence, we chose an ALD of 80 nm and opened the vias to the metal contacts by RIE. These ALD films had distinct advantages for the electrical results that will be discussed later.

## Experimental results

The separate customer designs were kept on individual dies but were merged into a master design together with our own test dies spread throughout the wafer. Figure 2a shows a photograph of a fabricated wafer of MPW run 1 (blurred to protect the confidentiality of the designs). Figure 2b, c show schematics of the MPW run 1 and run 3 masks, respectively. The green fields represent the 13 test dies, which were identical for both runs. Each test die contains 10 devices for four-point cross-bridge field-effect measurements and 10 devices for transfer length measurements (TLM), as shown in the optical microscopy images in Fig. 2d, e, respectively. Four-point measurements of

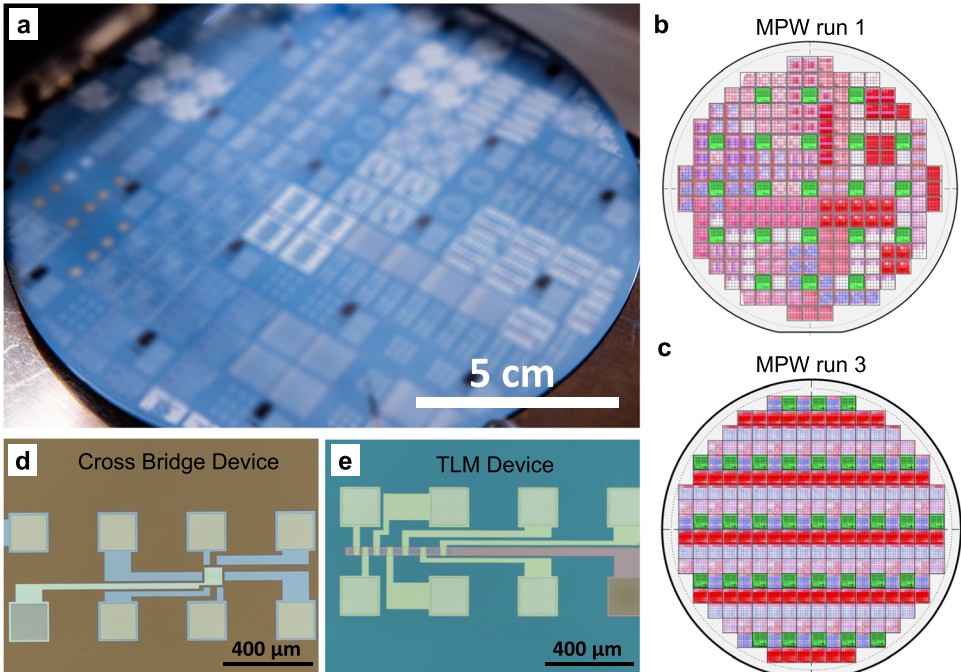

**Fig. 2 | Wafer fabricated in the Multi project wafer (MPW) runs and test devices.**
**a** Photograph of the 150 mm wafer with all the customer designs for MPW run 1.
Each die has a size of $1 \times 1\,cm^2$. Panel a is published with permission from the
Rightsholders. The panel has been blurred to protect the confidentiality of the
customer designs. **b** Schematic of the mask design (200 mm) for MPW run 1; the
green dies are the AMO test dies. **c** Schematic of the mask design (200 mm) for
MPW run 3. The green dies are the AMO test dies. **d** Optical microscopy image of
one cross-bridge device of MPW run 3 for four tip measurements. **e** Optical
microscopy image of one Transfer length method (TLM) device from MPW run 1.

the cross-bridge structures were used to extract the graphene sheet
resistance and mobility, while the TLM structures were used to
determine the contact resistance.

The specified target values for graphene mobility, Dirac voltage,
$R_{SH}$ and $R_C$ supplied to the customers are shown in Table 1, along with
the achieved results for both runs. The experimental data are based on
130 devices for MPW run 1 and 170 devices for run 3. The graphene
mobility is the median of the maximum hole mobility in all test devices
per run, extracted in the forward sweep direction. The values of the
sheet resistance (determined by 4-point measurements), the contact
resistivity (determined by TLM measurements), and the Dirac voltage
are also the median values of all test devices per run. The yield was
defined as the number of working test devices divided by the total
number of devices. The details of the electrical measurements are
discussed in more detail below.

Prior to the electrical characterization of the test devices, which
was obviously only possible after the completion of the wafers, we
conducted Raman spectroscopy, optical microscopy, atomic force
microscopy (AFM), and scanning electron microscopy (SEM) to
monitor and inspect the different fabrication steps. In particular,
Raman spectroscopy was performed statistically on the customer
wafers after graphene transfer and after fabrication. Raman maps were
measured at the center and edge of the wafers (see Methods and
Supplementary Note 1). For MPW run 1, the D peak at approximately
1350 cm$^{-1}$ is very small, indicating high graphene quality (Supplemen-
tary Fig. 1.1)[33]. More importantly, there is no noticeable change in the D
peak intensity before and after fabrication. For MPW run 3, on the
other hand, there is an emergence of additional peaks in the vicinity of
the D and G peaks as well as an increase in the D peak after device
fabrication (Supplementary Fig. 1.1). The Raman spectra for MPW run 1
were measured on devices without encapsulation, while the encapsu-
lation was present for the measurements of MPW run 3, but the main
cause for these additional peaks in MPW run 3 is the carbonization of
the photoresist residue on the graphene during the encapsulation of

the graphene by ALD, which was performed at a process temperature
of 300 °C. The $I_D/I_G$ increases from 0.04 to 0.07 after fabrication for
MPW run 1 and from 0.02 to 0.35 for MPW run 3 (Supplementary
Fig. 1.3a). Only encapsulation by ALD caused a significant increase in
the D peak (Supplementary Fig. 1.2). However, when fitting the over-
lapping peaks in this region of the spectrum, it is difficult to differ-
entiate between the D peak of graphene and the peaks attributed to
amorphous carbon. Therefore, it is quite possible that we are over-
estimating the amplitude of the D peak. The Raman analysis is dis-
cussed in more detail in Supplementary Note 1.

The electrical properties of the graphene were assessed by drain
current ($I_d$) versus gate voltage ($I_g$) measurements (transfer char-
acteristics) on test structures for both MPW runs. All cross-bridge,
four-point field-effect (Fig. 2d) and TLM (Fig. 2e) measurements were
performed under ambient conditions and at room temperature with
an automated Cascade probe station. Even though MPW run 1 offered
the option of opening the graphene channel, the test structures
measured were encapsulated by 200 nm of $Al_2O_3$, which was suffi-
ciently high for a stable encapsulation[34].

The transfer curves of all working devices of both MPW runs are
plotted for both the forward and backward sweep directions in
Fig. 3a, b. The sweep range of the gate voltage was $-40$ to $+40$ V for
MPW run 1 and $-10$ to $+10$ V for MPW run 3, while the source-drain
bias voltage was $V_{DS} = 100$ mV in both cases. A large sweep range of
80 V in MPW run 1 (Fig. 3) was necessary to capture the charge
neutrality point in both the forward and reverse sweep directions.
This was possible because of the rather large $Al_2O_3$ thickness of
75 nm. We observed a large hysteresis (37 V) for this sample due to a
high level of charge traps in the e-beam-evaporated $Al_2O_3$[35]. MPW run
3 had a much smaller hysteresis of 0.7 V. This is in line with the
typically much higher quality of ALD $Al_2O_3$[36], although an additional
reason is the smaller sweep range in MPW run 3 and a thinner gate
oxide. In fact, the equivalent oxide thickness (EOT) is 40 and 21 nm
for the devices in MPW runs 1 and 3, respectively. Figure 3c, d show

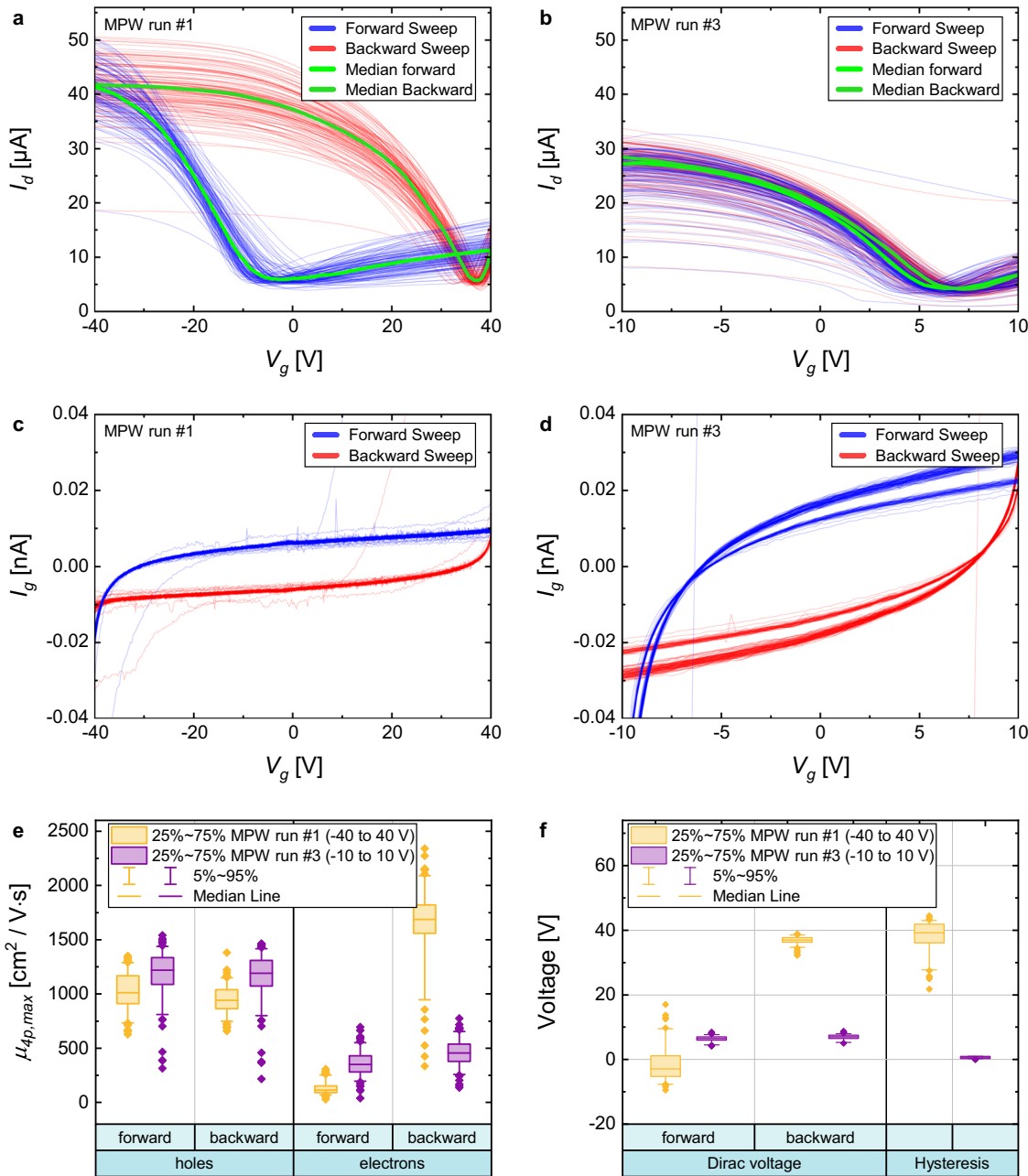

**Fig. 3 | Electrical characterization of the test devices for MPW runs 1 (130 devices) and 3 (170 devices). a** Transfer curves (drain current $I_d$ versus gate voltage $V_g$) for the MPW run 1 devices (sweep range from − 40 V to + 40 V). **b** Transfer curves for the MPW run 3 devices (sweep range from − 10 V to + 10 V). **c** Gate currents $I_g$ for MPW run 1. **d** Gate currents for MPW run 3. **e** Box plots showing the field effect mobility $\mu_{4P,max}$ of each device measured with 4 probes for both MPW runs. **f** Dirac voltages (forward and backward) and hysteresis for both MPW runs.

gate currents for MPW runs 1 and 3 below 50 pA, respectively. There is a slightly larger gate current in MPW run 3, which is related to the thinner dielectric film, but the transfer curves in Fig. 3a, b are clearly not dominated by this low gate leakage. The distribution of the Dirac points (forward and backward) and the hysteresis for both runs are shown in Fig. 3e as box plots confirming the good quality of the encapsulation in the MPW run 3.

The graphene field effect mobility across the wafers was extracted from the sheet resistance measured in the four-probe configuration on the cross-bridge structures[37]. We used a dielectric constant of $Al_2O_3$ of 7.3, which was calculated from capacitance-voltage (CV) measurements of MOS capacitors fabricated separately (see Supplementary Note 2). The channel lengths and widths of $L = 60\,\mu m$ and $W = 20\,\mu m$ were identical for both runs. We determined

the maximum hole and electron mobility in each sweep direction for all devices in both MPW runs. The distribution is shown as a box plot in Fig. 3f. The median field effect mobilities in the forward sweep direction are given in Table 1: 1023 cm²/Vs for MPW run 1 and 1185 cm²/Vs for MPW run 3. These values are the mean values calculated using the maximum of the transconductance method. Figure 4a, b show maps of the field effect mobilities of the test dies for MPW run 1 and MPW run 3, respectively. The values of each die correspond to the median of the maximum hole mobility in the forward direction for the working devices of each die. Figure 4c, d show a wafer map for the yield (working devices per die). The total yield for MPW run 1 was 94 % (122 of 130 devices), and that for MWP run 3 was 87 % (148 of 170 devices). A device was considered defective if the channel resistance exceeded 1 MΩ or the gate current

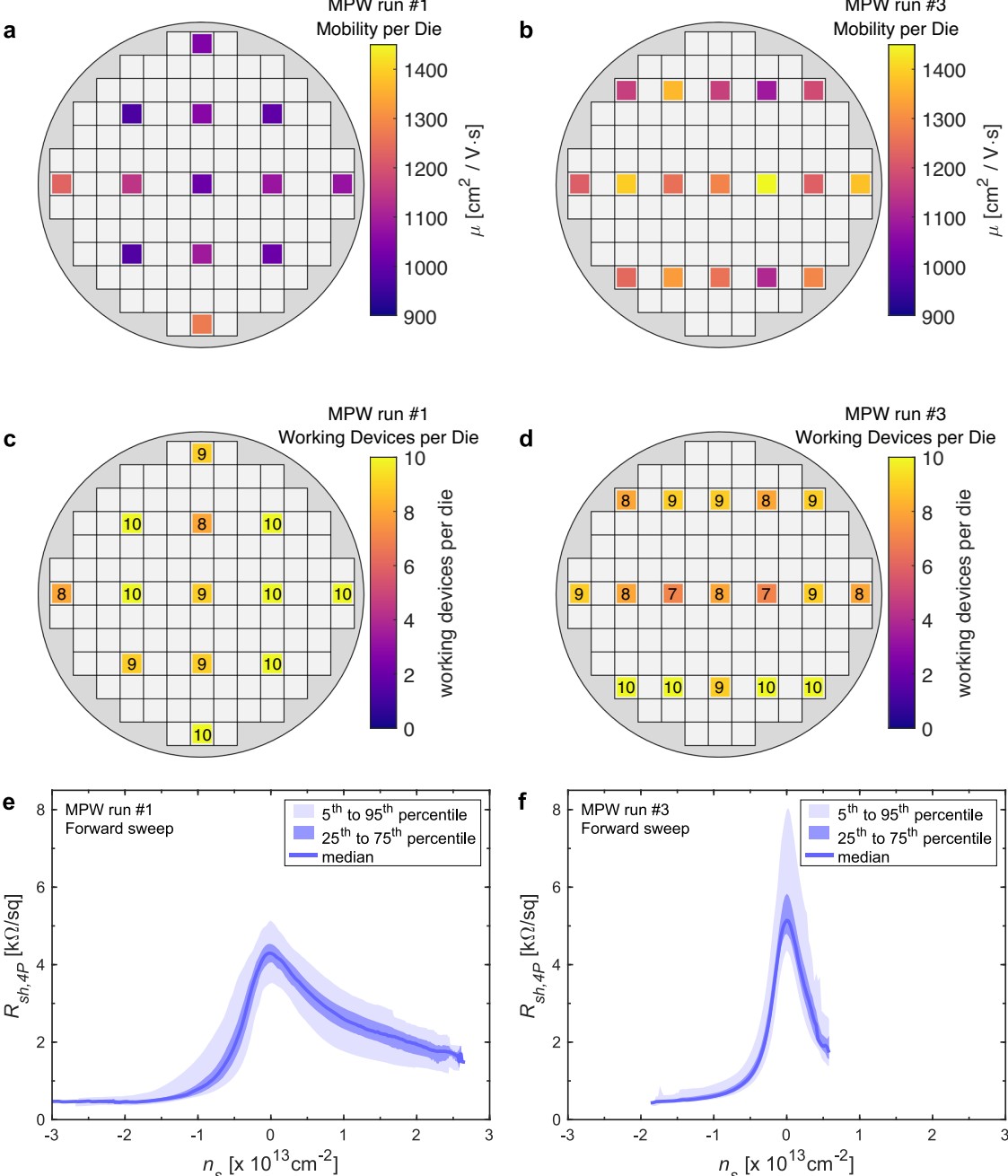

**Fig. 4 | Electrical characterization of the test devices for MPW run 1 and MPW run 3. a** Mobility map for MPW run 1 and (**b**) MPW run 3. The values represent the median value of the maximum four-probe mobility of all devices within the die. **c** Map with the number of working devices per die (there are 10 devices per die) for MPW run 1 (total wafer yield: 94%) from 130 devices measured. **d** Map with the number of working devices per die for MPW run 3 (total wafer yield: 87 %) from 170 devices measured. Each die has 10 devices. **e** Sheet resistance for the MPW run 1 wafer (forward sweep). **f** Sheet resistance for the MPW run 3 wafer (forward sweep).

exceeded 100 nA, although, for these runs, the high channel resistance due to defective graphene layers was the only cause of failure. Figure 4e, f shows the sheet resistance calculated from the four-point measurements for both runs plotted over the carrier concentration $n_s$. The range of $n_s$ is different for the two cases because it depends on the sweep range, the dielectric thickness, and the Dirac voltage. The target values are conservative compared to the literature and the general expectations those published values raise. When they were defined before the MPW runs, we not only considered previous experiments performed at AMO and other 2D-EPL partners on a wafer scale but also the fact that the MPW runs needed to provide flexible device design options that could impose non-ideal process flows.

The contact resistances were calculated from the measurements of the TLM structures. On the MPW run 1 wafer, the first five channels with lengths of 10, 20, 30, 40, and 50 μm and a channel width of 20 μm were measured (compare Fig. 2d). The data for the MPW run 3 wafer include all 6 channel lengths up to 60 μm. For each device, the sheet resistances and the contact resistances were extracted from the linear fit of the total resistance of each channel plotted over the different channel lengths, i.e., its slope ($R_{SH}/W$) and the intercept with the y-axis ($2R_C$), respectively. The median values of the extracted sheet resistance are the same as the sheet resistance extracted from the four probe measurements (see Table 1). The contact resistivities extracted at a carrier density of $n_s = -10 \times 10^{12}\,cm^{-2}$ are 0.8 and 1.5 kΩ·μm for MPW runs 1 and 3, respectively. At the charge neutrality

point, the values are 3.5 and 5.3 kΩ·μm for runs 1 and 3, respectively. Three of these values are slightly greater than the target values (see Table 1); however, it is difficult to determine the actual values, especially at the CNP, because the uncertainty of these values is very high, with sigma values larger than 5 kΩ·μm. This is not surprising because the TLM method, in principle, suffers from variations in contact and sheet resistance, which can lead to substantial resistance uncertainties in the extracted values[38,39]. This is particularly problematic in graphene devices because the sheet resistance of graphene grown by CVD is generally nonuniform due to the presence of grain boundaries, wrinkles and other defects such as mechanical tears and cracks[40,41] and the fabrication process is not free of residues[26,27]. Furthermore, the contact resistance depends on the carrier density in graphene, which can cause additional errors and even negative values in the extraction of the contact resistance (for additional information and discussion, see Supplementary Note 3)[21,24,42–51]. To determine the contact resistance with more precision, Kelvin probe microscopy could be used[52].

## Discussion

We encountered several specific technical and procedural challenges in our efforts to ramp up large-scale graphene device fabrication in these MPW runs, which went beyond the general issues described in the introduction. One difficulty originated in the PDK for MPW run 1, which was very loosely defined to reach as many stakeholders as possible. While this goal was achieved, additional design engineering and discussions with customers were required to enable multiple customer-specific applications. One of the main consequences was to use $Al_2O_3$ as both a back-gate dielectric and a top-encapsulation (see "Methods" section), which proved impossible to integrate in a conventional top-down process flow with wet etching or RIE. The solution was a lift-off process with e-beam evaporated $Al_2O_3$ (200 nm) of much lower quality than the ALD material originally foreseen. In addition, cracks formed in the encapsulation layer on top of the graphene but not on the other surfaces. These cracks extended through the entire film thickness down to the graphene (see details in Supplementary Note 4). In some cases, the encapsulation layer and the underlying graphene were completely delaminated, especially in large graphene areas of several hundred micrometers. This behavior is presumably caused by the low adhesion of graphene[53] to the substrate in conjunction with sufficiently high residual tensile stress in the evaporated encapsulation layer. In contrast, no cracks or delamination were observed in MPW run 3, which included only devices encapsulated by 80 nm $Al_2O_3$ by ALD.

Another challenge was the presence of resist residue, a well-known yet largely unsolved issue in graphene integration[10,14–17]. This known problem was particularly critical in MPW run 1 because there were many requests for graphene devices for biosensing applications, where a clean graphene surface is crucial. A solution beyond our standard process flow was needed, which required three lithography steps after graphene transfer and an additional coating with a protective resist before dicing the wafers. The amount of residue depends on the resist and process details in each step. The most crucial step in this regard is graphene patterning because the resist is exposed to plasma during the RIE of graphene, which can lead to resist crosslinking, making it extremely difficult to remove without damaging the graphene[17]. Some of the particular issues involving the resist residue are discussed in Supplemental Note 1.

Etching of the dielectric with RIE to create vias to the bottom gate or to the bottom contacts is one of the fabrication steps before the transfer of graphene, and it can also lead to crosslinked resist residue. Although more rigorous cleaning procedures can be applied when cleaning $Al_2O_3$ dielectrics than when cleaning graphene, some effective methods, such as piranha solution, are not possible because they also etch $Al_2O_3$.

The most important lesson learned from the first MPW run was that a tighter definition of the PDK would be beneficial. The flexibility of the initial PDK led to quite unexpected requests where customers creatively used some of the available layers for unintended purposes. While this made the run attractive to a larger group of stakeholders, it also led to significant technical difficulties in realizing all designs with different purposes and dimensions in the same run. Therefore, MPW run 3 and beyond will have a limited device library in the PDK, even if this means that fewer customers can be served with each run. These limitations will also apply to device dimensions, as controlling the graphene size is crucial for avoiding delamination. The PDK for MPW run 3 already solved many of the original issues with its stricter design rules.

The European Commission's 2D-Experimental Pilot Line project provided us with the opportunity to carry out multi-project wafer runs for graphene devices in 2022 and 2023, with 14 and 5 customers in MPW runs 1 and 3 of the 2D-EPL, respectively. We consider the wafer runs to be successful because the device yield and performance met or exceeded the initial specifications across the wafers. In addition, all customers were served with little or no delay. The learnings of MPW run 1 helped improve MPW run 3, which was finished without delays. The decrease in the number of orders and customers can be considered a consequence of the more limited freedom of chip design in the PDK. In addition, there was not much time between the delivery of the first batch of dies and the next run. This meant that several customers were still analyzing results when the new designs were due, although one returning customer can be seen as a positive result. Finally, several publications are expected with the dies fabricated in the 2D-EPL, which thus serves its purpose as an emerging platform for graphene device fabrication at a reliable demonstration level.

## Methods
### Device fabrication

The devices from MPW runs 1 and 3 were fabricated according to the workflows depicted in Fig. 1a and b, respectively. The workflow for both runs consists of six lithography layers. For all layers, a SüSS MA8 mask aligner and a mask with a 200 mm wafer size were used for contact lithography.

The metal contacts were deposited by e-beam evaporation (Pfeiffer evaporator in MPW run 1 and FHR evaporator in MPW run 3) followed by a lift-off process. The back gate (Layer 1) consisted of a layer of Pd/Ti (40 nm/5 nm) deposited on $SiO_2$ (90 nm)/Si. The $SiO_2$ layers were thermally grown on Si wafers with p-doping of 1–10 Ω·cm for MPW run 1 and on intrinsic, highly resistive (5 kΩ·cm) Si wafers for MPW run 3.

The next step was dielectric deposition and etching of the back gate pad vias (Layer 2). The dielectric was $Al_2O_3$ deposited by Plasma Enhanced Atomic Layer Deposition (PEALD) using trimethylaluminium (TMA) and oxygen plasma as precursors in an Oxford Instruments ALD FlexAL system. The thickness of the dielectric layer was 75 nm for MPW run 1 and 40 nm for run 3. The equivalent oxide thickness (EOT) is 40 and 21 nm for the devices in MPW runs 1 and 3, respectively. Vias were etched by reactive ion etching (RIE) in a Plasmalab System 100 from Oxford Instruments to access the back gate contact.

Layer 3 served different purposes for the two runs. In MPW run 1, this layer was the bottom part of the sandwich contacts used for this run. The bottom contacts were made from Pd/Ti (30 nm/5 nm) and fabricated by lithography, e-beam evaporation, and a lift-off process in the same way as the back gate (layer 1). In MPW run 3, layer 3 was the fabrication of 25 nm thick Ni adhesion pads underneath the contact pads of the top contacts using lithography, deposition and lift-off. The Ni was deposited by magnetron sputtering using a von Ardenne sputtering tool (Cluster system CS 730 S).

After this step, the wafers were sent from AMO to Graphenea for the transfer of graphene. Monolayer graphene grown on copper (Cu)

foil was transferred using a semi-dry transfer process. To accomplish this goal, graphene was initially protected by a carrier polymer to provide integrity during handling. Then, the Cu foil used was chemically etched away with a non-Fe-based solution, and the graphene layer was subsequently cleaned with different solutions of acids and distilled water. Once the graphene layer was clean, it was stamped on the target substrate, which was pretreated with gentle ozone activation of the surface to improve graphene adhesion. The stamping process was carried out using a semi-automatic system that controlled the pressure, residence time and temperature during stamping. The last step of graphene transfer consists of polymer removal with different solvents. For the MPW run 3, AMO also realized in-house transfer using commercial graphene from the brandmark Grolltex for wet transfer tests.

The wafers were then sent back to AMO, where the graphene was patterned (Layer 4) using positive lithography, and RIE with an $O_2$ plasma. This caused crosslinking of the remaining resist. Attempts to remove the crosslinked resist after etching were made using $O_2$ plasma (low flow and low power) and solvents. The process steps for layer 4 were the same for both MPW runs.

The next step was the fabrication of the top contacts (Layer 5). This step was the same in both runs. It was the last step of the sandwich contact fabrication for MPW run 1, while it was the only contact used for the devices made in MPW run 3. The step consisted of lithography followed by 40 nm evaporation of Pd and lift-off.

The encapsulation layer (Layer 6) was again different in the two MPW runs, as explained in the main text. For MPW run 1, $Al_2O_3$ was fabricated by e-beam evaporation (Pfeiffer evaporator) of 200 nm $Al_2O_3$. Because MPW run 1 offered an opening of graphene areas and this was only possible using a lift-off process (see main text), the fabrication sequence for this layer was lithography, evaporation and lift-off. The lift-off opened vias to the metal contacts and, where requested by the customer, opened the graphene areas for certain sensor devices. For MPW run 3, open graphene areas were not offered, so the $Al_2O_3$ deposition was made by ALD using a FlexAL Oxford tool in a sequential deposition of first thermal and then plasma ALD. The vias to the metal contacts were then etched by RIE in a Plasmalab System 100 from Oxford Instruments.

For the lithographies of Layers 4 and 5 (both runs) as well as Layer 6 in MPW run 1, manual processing was necessary, which made reproducibility difficult.

Different wafers were fabricated in both MPW runs because different customers had different requests. Here, we only reported the results for the main wafer for each run, although a number of other wafers were required for workflow development and for specific customer requests.

## Raman characterization

A Horiba XPlora Raman Spectrometer with a $100 \times$ microscope objective was used for Raman measurements. The laser power was set to 1 mW at an excitation wavelength of 532 nm. Raman spectroscopy was performed before graphene patterning as well as after fabrication for MPW run 1 and after each fabrication step for MPW run 3 to monitor potential changes in structural parameters such as defects and strain variations. $10 \times 10\ \mu m^2$ areas with 121 spectra each were measured at two wafer locations (center and edge). For each map, the accumulation time for each spectrum was increased until an acceptable signal-to-noise ratio > 20 was achieved. This led to measurements of 30 min or more per map, so it was not practical to measure more than two maps per fabrication step. The peaks of each spectrum were fitted by a Voigt function. A total of five peaks were used to fit the region of the D- and G-peaks to accommodate the appearance of amorphous carbon peaks after encapsulation in MPW run 3. The ratios between the D and G peak intensities, as well as the FWHM of the 2D peak, were determined, and histograms were created for the wafer. For the MPW run 3, the Raman spectra were acquired after each fabrication step, and the goal was to

control the modifications that the workflow may have caused in the graphene and compare this with the electrical results.

## Data availability

Relevant data supporting the key findings of this study are available within the article and the Supplementary Information file. All raw data generated during the current study are available from the corresponding authors upon request.

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

## Acknowledgements

This project received funding from the European Union's Horizon 2020 Research and Innovation Program under the Graphene Flagship 2D Experimental Pilot Line (2D-EPL, Grant No. 952792) and from the European Union's Horizon Europe Research and Innovation Program under the 2D Pilot Line (2D-PL, Grant No. 101189797).

## Author contributions

The experiments were conceived by B.C., M.O., D.N., G.R., Z.W., and M.C.L. The fabrication of the wafers was carried out by B.C., M.O., B.R., B.Ch., F.S., L.E., and S.L. Graphene was synthesized and transferred by A.M., A.C., and A.Z. The electrical measurements were performed by B.C., S.L.S., and E.R. Raman mappings were obtained by B.C. and M.O. AFM images were obtained by B.C. SEM images were obtained by M.O. Communication with customers was performed by G.R., B.C., L.E., and Z.W. The data were analyzed by B.C., M.O., A.H., G.R., Z.W., E.R., and M.C.L. All authors collaborated on the interpretation of the experiments. The manuscript was written and revised by all. The work was supervised by Z.W. and M.C.L.

## Funding

## Competing interests

B.C., M.O., B.R., B.Ch., S.L., A.H., F.S., L.E., S.L.S, D.N., G.R., Z.W., and M.C.L. are employees of the non-profit company AMO GmbH, whereas A.M., A.C., and A.Z. are employees of Graphenea S.A. Both companies are partners in the 2D-EPL project and are working on the development and integration of electronic devices based on graphene and other 2D materials. E.R. has no competing interest.
