## [Peer Review file · Nature Communications]

Multi-project wafer runs for electronic graphene devices in the European 2D-Experimental Pilot Line project

Corresponding Author: Professor Max Lemme

Version 0:

Reviewer comments:

Reviewer #1

(Remarks to the Author)

The manuscript by Canto. et. al discussed the "2D Experimental Pilot Line" (2D-EPL) project, and report the experience of the first and third multi-project wafer (MPW) runs as an outcome of the 2D-EPL. The batch fabrication of graphene transistors have been reported, and the process is large the same with previous literatures. Although the fundamental innovation advances are limited, this article indeed first report the multi-project wafer runs with detailed and solid results. I feel such reports could greatly accelerate the large-scale fabrication of 2D electronics and associated lab-to-fab transition. Therefore, it represents a significant effort toward the commercialization of graphene-based electronic devices and serves as a valuable reference industry point of view. I would like to support its publication in Nature Communications, with some questions below.

1. How to determine whether the graphene device has reached a commercially viable stage, I would suggest the author to provide some discussion.
2. The wet transfer process will eventually become a critical process to limit the overall device uniformity, performance and the industry-compatibility. Since the manuscript is targeting on the commercialization of graphene-based electronic devices, could the author discuss more about this limiting point.
3. Where does the target values for the MPW runs in Table 1 come from. Also, the uniformity should also be an important factor and should be included as a target value
4. In some devices, bottom contact is used, which would normally yield a higher contact resistance. Could the author discuss what is this structure used for. Also, all devices in this manuscript have long channel length > 10 um, could the author discuss the potential for further scaling down device dimensions in large scale fabrication.

Reviewer #2

(Remarks to the Author)

The use of GFET-based MPW to address challenges in existing fabrication processes and explore the potential for commercialization is noteworthy. However, the demonstrated device performance, yield, and variation are not satisfactory, and the issues raised by the authors are somewhat general, lacking the detailed analysis and discussion necessary to provide deeper insight.

Other groups have already demonstrated wafer-scale integration of 2D materials, with more detailed statistical analysis, as well as in-depth discussions of the challenges and solutions. For instance, IMEC has successfully demonstrated 300 mm growth and integration of electronic devices, providing findings related to the integration issues of 2D materials. In comparison to these previous studies, this paper seems to lack statistical analysis of device performance, in-depth discussion of the issues, and proposed solutions for improvement.

More specifically, I have the following comments:

1. Performance Variation: As shown in Figures 3 and 4, the performance variation across the wafer is quite broad, with some devices not functioning at all. Additionally, no wafer-to-wafer variation data are provided. The authors need to analyze the differences between devices with high I_{on} , low I_{on} , and those that failed in order to discuss potential improvements in performance variation and yield. Furthermore, off-current levels and the on/off ratio are not shown.

2. Cracks in Encapsulation Layer: On page 11 and in Figure S3.1, cracks in the encapsulation layer on top of the graphene are observed through OM and AFM. A detailed analysis and discussion of why these cracks formed and how they can be resolved is necessary.

3. EOT Scaling: The EOT values of 40 nm and 21 nm are relatively high, which increases the VDD. Further EOT scaling, either by reducing the thickness of Al₂O₃ or by switching to higher-k materials like HfO₂, would be advantageous. Previous studies on FETs with 2D materials have already demonstrated EOTs below 5 nm, with some even achieving close to 1 nm.

4. Hysteresis: The hysteresis in MPW run 1, as shown in Figure 3a, is quite large (37 V). The authors attribute this to charge traps in the evaporated Al₂O₃ layer, but considering that the channel was opened for sensor applications, it seems more likely that the large hysteresis is due to adsorbed atoms on the graphene channel surface during the V_g sweep. The authors should check the hysteresis under ambient and vacuum conditions for comparison.

5. Contact Structure: MPW runs 1 and 3 use different contact structures, with run 1 (dual contact) showing better contact resistance than run 3 (top contact). The authors should explain why they changed the contact structure and why run 1 achieved better contact resistance than run 3. Additionally, the channel length used for TLM is relatively long, which may reduce the reliability of the extracted contact resistance. The authors should provide the TLM fitting results and conduct TLM measurements with shorter channel lengths (down to 100 nm or less).

6. Performance Benchmarking: The authors should benchmark their device performance against previous research on GFETs. How does the state-of-the-art GFET performance compare to the results presented in this paper?

Reviewer #3

(Remarks to the Author)

In this manuscript, Canto et al. demonstrate the first experimental pilot line for the prototype of wafer-scale graphene electronics as part of the European 2D-EPL project. They have provided MPW runs as a service to various customers, and this manuscript primarily discusses the experiences and results of these efforts. The work holds significant value in advancing the commercialization of graphene and related 2D materials, particularly in verifying compatibility with standard silicon technologies, thus paving the way for large-scale manufacturing. Furthermore, this manuscript could offer valuable lessons to the broader 2D electronics community, contributing to 2D materials' potential adoption as the next-generation electronic material. However, I have a few concerns that need to be addressed before recommending this work for publication in Nature Communications. Here are my key concerns and suggestions:

The primary issue lies in the low target values and performance metrics. The target values for mobility, average sheet resistance, Dirac voltage, and other key parameters seem undervalued. Do these target values align with the general expectations of the industry for 2D materials? Could the authors clarify the rationale behind setting such low target values?

In MPW run 1, the CNP values in the forward and backward directions differ significantly, with one displaying a negative CNP and the other showing a large positive value. What is the reason behind this stark contrast?

MPW run 3 shows some performance degradation compared to MPW run 1, with higher average sheet resistance and contact resistance. Could the authors explain this degradation in performance?

On page 8, line 151, the authors mention that while the defect density in the graphene itself is minimal, Raman spectroscopy reveals the presence of amorphous carbon at the interface of graphene. This is a critical factor that could significantly degrade the performance of GFETs. The manuscript would benefit from a more in-depth discussion on how the authors plan to address this issue.

In Figure 4, mobility values vary significantly depending on the wafer position, with dies at the wafer edge showing 20–30% higher mobility than those at the center. What could be the cause of this variation? Also, how can we further improve it? 20% variation is not acceptable in industry.

In addition to the die-to-die variation, a more thorough examination of wafer-to-wafer variation is required. The manuscript should provide statistically extracted performance values for at least 10 out of the 75 wafers processed during the 2D-EPL project.

The TLM method introduces substantial uncertainties in the extracted contact/sheet resistance values. Could the authors discuss alternative methods that may provide more reliable measurements? This discussion is crucial for offering potential customers reliable reference specifications.

Version 1:

Reviewer comments:

Reviewer #1

(Remarks to the Author)

The authors have carefully addressed my previous questions, and I would support for its publication.

Reviewer #2

(Remarks to the Author)

Thank you for the detailed response. While there are some limitations due to the restricted MPW runs and measurements, the work provides valuable insights into the lab-to-fab transition. Most questions and comments have been addressed well, and I believe the revisions are appropriately reflected.

Reviewer #3

(Remarks to the Author)

The authors have provided a detailed response and addressed the points raised by the three reviewers. I appreciate the effort they have made in incorporating additional data obtained during and after the MPW runs, as well as extending the discussions in their manuscript. I understand the limitations regarding further experiments due to the unavailability of the wafers, and I find their explanations both reasonable and sufficient. Based on their responses and the revisions made, I am happy to support the publication of the current manuscript in Nature Communications. I have no further concerns at this stage.

Point by point response letter

For the manuscript “Multi-Project Wafer Runs for Electronic Graphene Devices in the European Project “2D-Experimental Pilot Line”

Dear reviewers,

Thank you for the efficient review process and for dedicating your valuable time to evaluating our manuscript. We also thank you for providing constructive and critical comments and suggestions. We have addressed all your comments in a point-by-point response below. You can find your comments in *italic black* and our response in **blue**. In the manuscript, we highlighted all changes in **red**. The additions and changes in the manuscript are also quoted below for your reference (in **red**). We have added data obtained during and after the completion of the MPW runs to our manuscript according to your suggestion. However, we are unable to perform additional experiments as requested because the wafers were diced and delivered to our customers or used within other projects. We nevertheless hope that our answers, additional data, and extended discussions suffice to address the points raised and that our manuscript can now be accepted for publication in Nature Communications.

Best regards,

The authors

Reviewer #1:

The manuscript by Canto. et. al discussed the "2D Experimental Pilot Line" (2D-EPL) project, and report the experience of the first and third multi-project wafer (MPW) runs as an outcome of the 2D-EPL. The batch fabrication of graphene transistors have been reported, and the process is large the same with previous literatures. Although the fundamental innovation advances are limited, this article indeed first report the multi-project wafer runs with detailed and solid results. I feel such reports could greatly accelerate the large-scale fabrication of 2D electronics and associated lab-to-fab transition. Therefore, it represents a significant effort toward the commercialization of graphene-based electronic devices and serves as a valuable reference industry point of view. I would like to support is publication in Nature Communications, with some questions below.

We thank Reviewer #1 for their positive comments and support. This paper is indeed meant to report our experience in a process that contributes to a lab-to-fab transition, possibly guiding future works in this area. We are happy that this came across and that the community supports us. On a private note, we meanwhile had several requests to share our experience from people eager to learn how the 2D Pilot Line operates and what our experiences were, including invitations to speak in Korea at SKKU, Hong Kong at the IEEE EDTM tutorial, Duke Univ. at the Device Research Conference 2025, and many others. We can currently point to the preprint on Researchsquare (<https://doi.org/10.21203/rs.3.rs-5032996/v1>), but we think a published version will create even greater interest.

1. How to determine whether the graphene device has reached a commercially viable stage, I would suggest the author to provide some discussion.

This is an essential point for the community, and we thank you for your encouragement. There is still a long way to go before the fabrication of graphene devices reaches a commercially viable stage, i.e., before production is taken up by commercial semiconductor manufacturers whose graphene device products are purchased by commercial customers. The 2D-EPL and the MPW runs were intended as a first step to provide fabricated chips for market actors so that they could “test the waters” with a pilot line for research and pre-product development. It was an experiment to see if there were customers for such a concept, which meant the pilot line runs had to be very flexible and unrestrictive with respect to the permitted device design. It was also meant to demonstrate that wafer-scale fabrication is feasible to a certain extent and can withstand the scrutiny of these paying customers. Accordingly, the target parameters for mobility and yield shown (see Table 1 in the manuscript) were chosen with certain compromise, i.e., keeping in mind that our pilot line facilities are not commercial production lines, whose aim is to fabricate a large number of customer designs with strict process design kits. This was well achieved.

Nevertheless, many issues still need to be addressed that are beyond the work presented in our manuscript, such as extensive reliability and lifetime measurements, or application-specific packaging of products with graphene devices. Also, although both MPW runs described in our paper were performed within the 2D Experimental Pilot Line using processes that are very similar to industry standards, they are not as fixed or rigid as production processes, and, as such, they are not production-ready.

To clarify the 2D-EPL achievements to the readers more accurately and to transparently inform them of what it did and didn't (attempt to) achieve, we added some discussion about this topic in the Introduction. This paragraph also includes clarifications about the comment #3 (see below):

“If the target values were reached and the optical microscopy analysis showed an acceptable quality regarding the lift-off and etching, we considered the wafer ready for delivery. Table 1 was defined in the initial 2D-EPL project description as specifications for potential customers and published on the 2D-EPL website. Although higher values can and have been achieved, the specifications were set conservatively to ensure we could deliver what we promised to the customers. One parameter that was not included in the table is uniformity because the table was created for the benefit of the customers, who only ordered single chips from the MPW run, so uniformity was of secondary importance to them. Nevertheless, wafer scale uniformity is a very important parameter for industrial processes and was therefore measured and will be discussed in this section.”

2. The wet transfer process will eventually become a critical process to limit the overall device uniformity, performance and the industry-compatibility. Since the manuscript is targeting on the commercialization of graphene-based electronic devices, could the author discuss more about this limiting point.

We agree with your assessment of the likely limitations of wet transfer. In fact, one of the targets of the 2D-EPL is the development of industry-level process modules, including wafer-scale transfer. However, there is also a second aim of the 2D-EPL that targets the delivery of graphene-based devices to customers. These two goals cannot be achieved simultaneously because their timelines do not match, i.e., turn-key, production-ready tools and processes for wafer-scale transfer were not yet available for these MPW runs (and still aren't). Instead, the two goals were pursued in parallel. We chose the semi-dry transfer because it completed our wafer-scale fabrication scheme of graphene devices with sufficient maturity, allowed us to provide our R&D-orientated customers with dies for testing, and, hence, yielded the 2D-EPL targets for the MPW runs even without fully industry-compatible processing,

The current situation regarding the dry transfer within the 2D-EPL project is that IMEC and Suss Microtech developed dry and semi-dry transfer processes based on 300 mm wafer bonding techniques. This work is continued in the new 2D-PL (2D-Pilot Line) project, a follow-up project to the 2D-EPL. The 2D-PL expands this research to include other 2D materials such as TMDCs and further industrial partners like EVG (Austria), for example. The 2D-PL project will also offer MPW runs, including more industrially relevant processes as they become available, especially when offered by institutions higher along the TRL value chain, such as IMEC. To clarify this to the reader, we added some discussion about this topic in the Introduction:

“In fact, one of the targets of the 2D-EPL is the development of industry-level process modules. There is also a second aim of the 2D-EPL that targets the delivery of graphene-based devices to customers. These two goals cannot be achieved simultaneously because their timelines do not match, i.e., turn-key, production-ready tools and processes for wafer-scale transfer were not yet available for these MPW runs. Instead, the two goals were pursued in parallel. The second target, the availability of multi-project wafer runs (MPW runs) as a service, was one of the most challenging goals, which were offered by different 2D-EPL partners with suitable clean room facilities.”

3. Where does the target values for the MPW runs in Table 1 come from. Also, the uniformity should also be an important factor and should be included as a target value

The target values in Table 1 were defined in the initial 2D-EPL project description upon submission of the proposal to the competitive call by the European Commission. They were intended as specifications for potential customers and, once the project was selected for funding, published on the 2D-EPL website. Its values are based on previous results and experience at AMO GmbH and other 2D-EPL partners with wafer-scale processing. Although higher values can and have been achieved, the specifications were set conservatively to ensure we could deliver what was promised to the customers. In particular, we accounted for fabrication challenges due to potentially many different custom designs on the wafers, which, in fact, we faced in the end.

We agree that uniformity on a wafer scale is an essential and typical specification in semiconductor processing. It was not included as a specification because the table was created for the benefit of the customers, who only ordered single chips from the MPW run. Hence, they cared about the device specs, while wafer-scale uniformity was

something for AMO to deal with in-house: low uniformity would have meant low yield and, thus, higher fabrication cost. We did account for uniformity and yield in our internal calculations, which are confidential. In summary, although uniformity was not an important specification for the customers, and is therefore not included in Table 1, it was an important aspect of the MPW run experience. It was measured and the data is shown in Figure 4 in the manuscript.

To clarify the uniformity question to the readers, to connect this topic with the 2D-EPL achievements, and to discuss what the 2D-EPL did and didn't (attempt to) achieve, we add the following sentences to the Introduction (repeated here from topic #1).

“If the target values were reached and the optical microscopy analysis showed an acceptable quality regarding the lift-off and etching, we considered the wafer ready for delivery. Table 1 was defined in the initial 2D-EPL project description as specifications for potential customers and published on the 2D-EPL website. Although higher values can and have been achieved, the specifications were set conservatively to ensure we could deliver what we promised to the customers. One parameter that was not included in the table is uniformity because the table was created for the benefit of the customers, who only ordered single chips from the MPW run, so uniformity was of secondary importance to them. Nevertheless, wafer scale uniformity is a very important parameter for industrial processes and was therefore measured and will be discussed in this section.”

4. In some devices, bottom contact is used, which would normally yield a higher contact resistance. Could the author discuss what is this structure used for. Also, all devices in this manuscript have long channel length > 10 μm, could the author discuss the potential for further scaling down device dimensions in large scale fabrication.

We agree with your assessment of bottom contacts. However, we did not use bottom contacts in this work but sandwich contacts. These have been shown in the literature as a good option to minimize contact resistance¹. We explained the details in the manuscript in three different positions:

1. line 109: “In MPW run 1, the graphene was sandwiched between the bottom and top contacts, while in MPW run 3, only the top contacts were used.”
2. line 288: “In MPW run 1, this layer was the bottom part of the sandwich contacts used for this run.”
3. line 308: “It was the last step of the sandwich contact fabrication for MPW run 1, while it was the only contact used for the devices made in MPW run 3.” (We also included the reference cited above in this paragraph).

It is possible to scale down the channel length by optimizing the process and, for devices below 1 μm channel length, changing the lithography technique. The MPW runs did not include this option because changing from contact lithography to projection lithography or e-beam lithography would have made the individual dies much more expensive than the target costs for these MPW runs (see also response to Reviewers #2 and #3). Another important point to be considered is that the target application for the MPW runs was

graphene-based sensor devices, and for this application, downscaling to sub μm dimensions is usually not required nor desired.

Reviewer #2:

The use of GFET-based MPW to address challenges in existing fabrication processes and explore the potential for commercialization is noteworthy. However, the demonstrated device performance, yield, and variation are not satisfactory, and the issues raised by the authors are somewhat general, lacking the detailed analysis and discussion necessary to provide deeper insight.

Other groups have already demonstrated wafer-scale integration of 2D materials, with more detailed statistical analysis, as well as in-depth discussions of the challenges and solutions. For instance, IMEC has successfully demonstrated 300 mm growth and integration of electronic devices, providing findings related to the integration issues of 2D materials. In comparison to these previous studies, this paper seems to lack statistical analysis of device performance, in-depth discussion of the issues, and proposed solutions for improvement.

We thank Reviewer #2 for the critical and constructive comments. We would like to clarify the intention and scope of our work to address the comment about performance, yield and variation. The MPW runs are part of the 2D Experimental Pilot Line (2D-EPL), a research and development project funded by the European Commission. One goal of the project was to offer prototyping services for graphene devices on 150- and 200-mm wafers, while another goal was to develop automated process modules on 200- and 300-mm wafers and, ultimately, transfer this knowledge to the prototyping services and the semiconductor industry. The MPW runs obviously addressed the former. Thus, the goal was always to provide a flexible service to as many interested customers as possible with low-TRL technology. Our manuscript aims to explain the outcome and learnings from this experience. Our manuscript reports the results of our work, which never had the goal of reaching record device performance, yield, or low variability but to create an unprecedented flexible entry point to delivering devices and chips according to predefined specifications (see also answer 3 to Reviewer 1).

The two MPW runs described in our manuscript were two of the first three offered in the 2D-EPL. They were performed at AMO, a non-profit research institute without the infrastructure and tools to provide the state-of-the-art manufacturing available in large research centers such as IMEC or commercial foundries. Consequently, the paper is not about state-of-the-art integration of 2D materials in a 300 mm platform using industrially relevant processes. It rather reports on the successful (first) prototyping of wafers with many different custom designs at an affordable price under a deadline, achieved despite low manufacturing readiness levels of 2D materials and their process technology. Because the wafer runs were made for customers with strict deadlines, most chips were immediately delivered to the customer sites upon completion, where most of the performance analyses were carried out. The resulting data are typically confidential and cannot be included in the manuscript, although we know of several forthcoming R&D results based on the delivered chips.

We further agree that wafer-scale integration of 2D materials has been shown previously but with different intentions and purposes. IMEC, a partner in the 2D-EPL project, typically

carries out their research on the 200- and 300-mm wafer scale. This is their role in the value chain. Their analyses are typically automated and, therefore, extensive and concern certain aspects of a specific device type. Beyond conventional silicon MOSFETs, FinFETs, or nanosheet FETs, these recently include less mature technologies like 2D semiconductor-based FETs. However, Pilot line services are typically only offered if and when a technology reaches a certain maturity. This is typically accompanied by a detailed description in a process design kit (PDK) so that customers can design and order chips without understanding the full underlying technology. The MPW runs in such mature technologies are typically accessed in Europe through EUROPRATICE, and tape-out costs often amount to tens of thousands of Euros or more. Graphene and 2D materials-based devices do not have that maturity yet. Instead, the 2D EPL and the services described in the manuscript are meant to be an entry point toward such an ecosystem.

In summary, the manuscript describes the first 2D-EPL outcomes toward becoming a prototyping platform for research institutes and early adopters from industry. It is an essential step in building a manufacturing ecosystem, including specifications, PDKs and reliable delivery times. We have tried to make the goals more transparent in the updated introduction (repeated here from answer 2 to Reviewer #1):

“In fact, one of the targets of the 2D-EPL is the development of industry-level process modules. There is also a second aim of the 2D-EPL that targets the delivery of graphene-based devices to customers. These two goals cannot be achieved simultaneously because their timelines do not match, i.e., turn-key, production-ready tools and processes for wafer-scale transfer were not yet available for these MPW runs. Instead, the two goals were pursued in parallel. The second target, the availability of multi-project wafer runs (MPW runs) as a service, was one of the most challenging goals, which were offered by different 2D-EPL partners with suitable clean room facilities.”

More specifically, I have the following comments:

1. Performance Variation: As shown in Figures 3 and 4, the performance variation across the wafer is quite broad, with some devices not functioning at all. Additionally, no wafer-to-wafer variation data are provided. The authors need to analyze the differences between devices with high I_{on} , low I_{on} , and those that failed in order to discuss potential improvements in performance variation and yield. Furthermore, off-current levels and the on/off ratio are not shown.

We have divided the answer to this comment into several points:

1. Performance variation: We obviously agree that the yield was below 100%, with some devices not working at all. We explain this in our original manuscript in lines 192 to 195: “In MPW run1, the yield was 94% and in the MPW run 3 the yield was 87%. We considered a device defective if the channel resistance exceeded 1 M Ω or the gate current exceeded 100 nA. In these two MPW runs, the high channel resistance was almost exclusively due to defective graphene layers, most likely caused by the non-automated transfer”. (see also answer 2 to Reviewer #1).

2. Wafer-to-wafer variation: We have provided analyses of two wafers with different dielectric thicknesses, different encapsulation layers, and different contacts in the

manuscript. This led to different voltage sweep ranges to measure them. Hence, the data of the two wafers are not comparable. This is explained in the Methods section and in the Fabrication Workflow section of the manuscript. We also mention that although 75 wafers were processed in total, not all of them were measured. Most were used for tests or abandoned at some point of fabrication for various reasons (e.g., selected after a process variation, process flows hitting dead ends, etc.). Therefore, there is no large pool of comparable wafers available for a more thorough investigation. We also did not perform costly multi-wafer process runs just for the sake of obtaining more data, as this is an industry exercise for product development beyond the scope of this part of the 2D EPL project and a non-profit research institute like AMO. Hence, more detailed wafer-to-wafer variations are not available.

3. High I_{on} , low I_{on} , on/off ratio: These parameters are not relevant for graphene field effect transistors measured here, because graphene transistors cannot be switched off. Therefore, there is no I_{off} state. That is why there is no discussion about these parameters.

2. Cracks in Encapsulation Layer: On page 11 and in Figure S3.1, cracks in the encapsulation layer on top of the graphene are observed through OM and AFM. A detailed analysis and discussion of why these cracks formed and how they can be resolved is necessary.

We thank you for pointing out this aspect. We assume that the cracks were formed by the tension of the encapsulation film on the graphene in conjunction with the relatively low adhesion at the graphene interface². However, we regret that we cannot provide more information on this because the wafers were diced and shipped to the customers to meet delivery deadlines, and the customer's data is confidential (see also our reply above). Hence, no more studies could be conducted on these wafers regarding the cracks at that moment. Moreover, it is also impossible to do further experiments even on a related or repeated wafer run because the tool with which these films were grown was decommissioned shortly after the MPW run1 was completed. We added reference ² to the main text to address your concern. This was not an issue for the MPW run #3 because the deposition method for encapsulation was ALD instead of e-beam evaporation.

3. EOT Scaling: The EOT values of 40 nm and 21 nm are relatively high, which increases the VDD. Further EOT scaling, either by reducing the thickness of Al₂O₃ or by switching to higher-k materials like HfO₂, would be advantageous. Previous studies on FETs with 2D materials have already demonstrated EOTs below 5 nm, with some even achieving close to 1 nm.

We agree that the EOT values are quite high, particularly considering modern CMOS device requirements. We have used Al₂O₃ for different reasons. (1), We had a well-established atomic layer deposition process available in the AMO clean room, with high homogeneity and known interaction with graphene devices, including the resulting mobility. This was more important than achieving particularly low EOT values for the targeted customer runs, especially as the development of other dielectrics would have posed severe risks to the timeline of the MPW runs. We politely remind the Reviewer that

the targeted applications of our customers for these two MPW runs were not integrated circuits with low operating voltages, so very low EOTs were not necessary. Also, some clients required large area devices and also needed to ensure that the gate dielectrics were stable even when exposing the devices to different chemicals for functionalization. Therefore, a high EOT became necessary.

In addition, lacking a damascene process at AMO, the back gate metallization layer was realized with a lift-off process. This can sometimes result in leakage currents or breakdown due to metallic residues at the edges. We, therefore, conservatively decided to use the relatively thick dielectrics.

In summary, since the target applications were not electronics but sensing, scaling of the dielectrics was not critical for our customers. Therefore, this was also not the focus of the MPW runs and this paper.

4. Hysteresis: The hysteresis in MPW run 1, as shown in Figure 3a, is quite large (37 V). The authors attribute this to charge traps in the evaporated Al₂O₃ layer, but considering that the channel was opened for sensor applications, it seems more likely that the large hysteresis is due to adsorbed atoms on the graphene channel surface during the V_g sweep. The authors should check the hysteresis under ambient and vacuum conditions for comparison.

We agree with your assessment based on the assumption that the channels had been open to the ambient air, as was offered to some of the customers. However, this is a misunderstanding, probably caused by Figure 1, which shows the channel opened. In fact, the hysteretic electrical data shown in the paper is from GFETs on our test dies. These were indeed encapsulated. We apologize for this misunderstanding and have tried to clarify this by adding the following information to the end of page 9:

“Even though MPW run 1 offered the option of opening the graphene channel, the test structures measured were encapsulated by 200 nm of Al₂O₃, which was sufficiently high for a stable encapsulation.³”

This is also why we did not measure the GFETS in a vacuum.

5. Contact Structure: MPW runs 1 and 3 use different contact structures, with run 1 (dual contact) showing better contact resistance than run 3 (top contact). The authors should explain why they changed the contact structure and why run 1 achieved better contact resistance than run 3. Additionally, the channel length used for TLM is relatively long, which may reduce the reliability of the extracted contact resistance. The authors should provide the TLM fitting results and conduct TLM measurements with shorter channel lengths (down to 100 nm or less).

The contacts from MPW run 1 indeed provide higher performance. We expected this both from the literature¹ and our previous experiments. Nevertheless, the change was necessitated by logistic reasons in the clean room, i.e. the lack of availability of a specific evaporator and its subsequent decommissioning.

We agree with your statement regarding the reliability of the TLM method, as also discussed in the responses to Reviewer #3. However, making shorter channels would have required using either projection lithography or e-beam lithography. Here, we had the constraint to use contact lithography to make this process accessible to customers willing to pay only a specific price for chips made in the MPW runs of the 2D-EPL. Fabricating channel lengths of 100 nm or less by e-beam lithography would have been prohibitively expensive for these customers, especially those who only required very large structures, e.g. for sensing applications (see also above and response to Reviewer #1).

We further agree that TLM is not a perfect method for extracting contact resistances, especially when using large channel lengths and thin channel materials like graphene^{4,5}. Even small fluctuations in the contact resistance R_c and the sheet resistance R_{sh} , which often occur in devices with CVD-grown graphene, can lead to considerable uncertainties in the values extracted from the TLM fits, including negative (unrealistic) R_c values. We already tried to explain this general dilemma in our original manuscript as follows:

“Three of these values are slightly greater than the target values (see Table 1); however, it is difficult to determine the actual values, especially at the CNP, because the uncertainty of these values is very high, with sigma values larger than $5 \text{ k}\Omega\text{-}\mu\text{m}$. This is not surprising because the TLM method, in principle, suffers from variations in contact and sheet resistance, which can lead to substantial resistance uncertainties in the extracted values.^{6,7} This is particularly problematic in graphene devices because the sheet resistance of graphene grown by CVD is generally nonuniform due to the presence of grain boundaries, wrinkles and other defects such as mechanical tears and cracks^{8,9}, and the fabrication process is not free of residues.^{10,11} Furthermore, the contact resistance depends on the carrier density in graphene, which can cause additional errors and even negative values in the extraction of the contact resistance (for additional information and discussion, see Supplementary Information S3).^{4,12–22}

During the preparation of the original manuscript, we discussed how to present our TLM data best. Since we (expectedly) obtained also negative values that have no physical meaning, we decided to include only the positive values in the manuscript. These were also reported to the MPW run customers. This means that negative values of R_c , caused by large uncertainties of the fit, were deleted. This leads to a conservative median value of R_c that is too high and to sigma values that are too low. Figure RL1 shows the entire set of TLM plots and histograms of R_c and R_c plotted over n_s for both MPW runs. The median R_c values of MPW runs 1 and 2 are 438 and 619 $\text{Ohm}\ \mu\text{m}$, respectively, but the sigma values are so high (4918 and 6609 $\text{Ohm}\ \mu\text{m}$) that the median values are not trustworthy. We added a section in the Supplementary Material (S3) to explain this. There, the plots are shown with and without negative contact resistance for transparency, accompanied by the following added discussion:

S3 contact resistance calculation

“As mentioned in the main text, the TLM method is not necessarily a reliable method for the extraction of the contact resistance. For the TLM method to be reliable, certain conditions need to be met, one of them being that R_c , R_{sh} and the channel width w remain constant for all channels. When this is not the case, the resulting extraction values of R_c

and R_{sh} from the linear fit are no longer trustworthy. Even small fluctuations in the actual values can lead to extremely large errors in the extracted values.^{6,7} In graphene, these fluctuations can be particularly large, for example due to mechanical damage (cracks or holes) in the graphene, which leads to high variability of the channel resistances.

In particular, the reliability of the contact resistance extraction at the charge neutrality point (CNP) is heavily affected by the presence of a p-p⁺ step at the metal-graphene contact, which introduces an additional series junction resistance (R_{JUN}) that is not accounted for in the TLM model. Therefore, the incorrect modeling leads to a large number of negative values that have no physical meaning. For this reason, the values extracted at high negative bias are most reliable, where the effect of R_{JUN} is negligible²³.

Shorter channel lengths (for example less than 100 nm) can alleviate this problem, however, making shorter channels would have required using either stepper lithography or e-beam lithography. Here, we had the constraint to use contact lithography to make this process accessible to customers willing to pay only a specific price for chips made in a MPW run of the 2D -EPL. Fabricating channel lengths of 100 nm or less by e-beam lithography would have been prohibitively expensive for these customers, especially those who only required very large structures, e.g. for sensing applications.

During the preparation of the original manuscript, we discussed how to present our TLM data in a more precise way. Since we (expectedly) obtained also negative values due the imprecision of the TLM method, we decided to include only the positive values in the manuscript. These were also reported to the MPW run customers. This means that negative values of R_c , caused by large uncertainties of the fit, were deleted. This leads to a conservative median value of R_c that is too high and to sigma values that are too low. Figure S3.1 shows the entire set of TLM plots and histograms of R_c and R_c plotted over ns for both MPW runs with negative values. The median R_c values of MPW runs 1 and 3 are 438 and 619 Ohm μm , respectively, but the sigma values are so high (4918 and 6609 Ohm μm) that the median values are not trustworthy. More values can be seen in S13. Figure S3.2 shows the same data set of TLM plots with the negative data ignored (the data set used in Table 1). “

An alternative for calculating the contact resistance is to use Kelvin Probe Microscopy, but we have no access to a tool to do such measurements automatically and on a wafer scale. We nevertheless added this brief discussion in the manuscript:

“To determine the contact resistance with more precision, Kelvin probe microscopy could be used.²⁴”.

Figure RL1: TLM fittings extracted from the TLM measurements of the two wafers for two different n_s . Histograms of the extracted R_c and R_c over n_s , including positive and negative values.

6. Performance Benchmarking: The authors should benchmark their device performance against previous research on GFETs. How does the state-of-the-art GFET performance compare to the results presented in this paper?

This is, admittedly, a tricky question for us. There are obviously champion devices in the literature with carrier mobilities exceeding $10.000 \text{ cm}^2/\text{Vs}$, but manufacturing a few champion devices was never a target of the 2D-EPL work (and our manuscript). The mobility is, therefore, low compared to those records in the literature. Our goal, instead, was to deliver prototype devices with clearly defined specifications for many customers with different target applications in a unified process. We have met this goal and, in our opinion, this represents a state-of-the-art of a different kind. (see also answers to Reviewer #1). We would, therefore, prefer to refrain from benchmarking (which would seem, to be frank, like comparing apples and oranges).

Reviewer #3:

In this manuscript, Canto et al. demonstrate the first experimental pilot line for the prototype of wafer-scale graphene electronics as part of the european 2D-EPL project. They have provided MPW runs as a service to various customers, and this manuscript primarily discusses the experiences and results of these efforts. The work holds significant value in advancing the commercialization of graphene and related 2D materials, particularly in verifying compatibility with standard silicon technologies, thus paving the way for large-scale manufacturing. Furthermore, this manuscript could offer valuable lessons to the broader 2D electronics community, contributing to 2D materials' potential adoption as the next-generation electronic material. However, I have a few concerns that need to be addressed before recommending this work for publication in Nature Communications. Here are my key concerns and suggestions:

We thank Reviewer #3 for the positive comments and the general support, and especially for recognizing that the manuscript could offer valuable lessons to the 2D electronics community.

The primary issue lies in the low target values and performance metrics. The target values for mobility, average sheet resistance, dirac voltage, and other key parameters seem undervalued. Do these target values align with the general expectations of the industry for 2D materials? Could the authors clarify the rationale behind setting such low target values?

We agree that the target values, as summarized in Table 1 in our manuscript, are conservative compared to many reports in the literature and the general expectations those published values raise. Table 1 was defined in the initial 2D-EPL project description upon submission of the proposal to the competitive call by the European Commission. The values were intended as specifications for potential customers and, once the project was selected for funding, published on the 2D-EPL website. They are based on previous results and experience at AMO GmbH and other 2D-EPL partners with wafer-scale processing, and the fact that the MPW runs needed to have flexible device design options that could impose non-ideal process flows. We expected to face a different situation from typical research and development, where certain (subsets of) parameters are optimized to achieve a specific performance target, which then may set a record of some type. Here, trust in and dependability on the specs were key and enabled the development of the first

prototype process design kits (see also our answer to your next question, which explains how customer requests led to non-ideal processing conditions). We have added the following sentence to the manuscript to explain our reasoning better:

“The target values are conservative compared to the literature and the general expectations those published values raise. When they were defined before the MPW runs, we not only considered previous experiments performed at AMO and other 2D-EPL partners on a wafer scale but also the fact that the MPW runs needed to provide flexible device design options that could impose non-ideal process flows.”

In MPW run 1, the CNP values in the forward and backward directions differ significantly, with one displaying a negative CNP and the other showing a large positive value. What is the reason behind this stark contrast?

The hysteresis in the current-voltage characteristics of MPW1 run is indeed very large. This is a direct consequence of the complex requirements of several customers. Their device requirements meant we had to use Al₂O₃ grown by electron beam evaporation for encapsulation instead of our typical atomic layer deposited material. We were able to use the latter in MPW run3, with much lower hysteresis but also with much less process flexibility. We expanded our original text in lines 171 and 174 that this hysteresis was from the charge traps of the Al₂O₃ and also the dielectric below the graphene as follows:

“We observed a large hysteresis (37 V) for this sample due to a high level of charge traps in the e-beam-evaporated Al₂O₃.²⁵ MPW run 3 had a much smaller hysteresis of 0.7 V. This is in line with the typically much higher quality of ALD Al₂O₃²⁶, although an additional reason is the smaller sweep range in MPW run 3 and a thinner gate oxide.”

MPW run 3 shows some performance degradation compared to MPW run 1, with higher average sheet resistance and contact resistance. Could the authors explain this degradation in performance?

The reason for the contact resistance degradation was the choice of top contacts (MPW run 3) instead of sandwich contacts (MPW run1). While this is a performance disadvantage, it made processing less complex and was a known compromise.

To clarify this, we included the following sentence in the manuscript in the section “fabrication workflow”:

“This was done to reduce complexity and accelerate processing, even though the contact resistance might be higher for top contacts than sandwich contacts.”

The increase in sheet resistance comes from the lower dielectric thickness which meant that we use smaller voltage sweep ranges in our electrical measurements compared to MPW run 1. This resulted in higher mobility than MPW run 1 but increased sheet resistance (i.e. lower carrier density).

In general, the fabrication processes were different for the two MPW runs for logistic purposes due to customer requests and feedback, and because the two runs had different objectives, as mentioned in lines 56, 57, 107 and 108 of the manuscript text:

“MPW run 1 was intended mainly for 57 graphene-based sensors, while MPW run 3 focused on graphene electronics.”

On page 8, line 151, the authors mention that while the defect density in the graphene itself is minimal, Raman spectroscopy reveals the presence of amorphous carbon at the interface of graphene. This is a critical factor that could significantly degrade the performance of GFETs. The manuscript would benefit from a more in-depth discussion on how the authors plan to address this issue.

There is a discussion in the Supplementary information about the detected amorphous Carbon, including an in-depth Raman analysis. Since this includes substantial technical information, we added this to the Supplementary information (S1 Raman Analysis). The main discussion points in the SI are as follows:

“These peaks can be associated with amorphous carbon (a-C peak).^{27,28} However, when fitting the overlapping peaks in this region of the spectrum, it is difficult to differentiate between the D peak of graphene and the peaks attributed to the amorphous carbon. Because of this, it is quite possible that we are overestimating the amplitude of the D peak. The main cause for these additional peaks in MPW run 3 is the deposition method of the encapsulation in conjunction with the presence of photoresist residue on the graphene. For MPW run 1, the encapsulation was deposited by e-beam evaporation and the wafer remained at a temperature of less than 50 °C. For MPW run 3, the encapsulation was deposited by ALD at a chuck temperature of 300 °C, which corresponds to a wafer temperature of around 200 °C. This is hot enough to burn the resist residue from the graphene patterning step. The reason we believe this is related to burned resist and not to the ALD process itself, will be discussed here. The encapsulation process that was used (more details in the Methods section of the main paper) is a combination of the thermal ALD and PEALD process at 300 °C. However, these peaks only appear if the graphene has been subjected to lithography and RIE prior to ALD, i.e. there is resist residue present on the graphene. For example, Figure S1.2b shows spectra for three different scenarios before and after encapsulation: first, the encapsulation by e-beam evaporation at less than 50 °C from MPW run 1, second, the encapsulation by ALD at 300 °C and third, the encapsulation of graphene by ALD at 300 °C on a test wafer without prior lithography and RIE on the graphene. It is clear in the spectra from Figure S1.2b that the a-C peaks only appear for the combination of higher temperatures and prior lithography, leading us to believe that burning of resist residue is responsible for the appearance of these peaks. We were also able to repeat these results by simply heating samples of graphene with and without photolithography in vacuum, i.e. without deposition by ALD for temperatures between 100 and 300 °C (not shown here), confirming that ALD itself is not responsible for these peaks.”

For your information, we have another work in progress exploring different methods for cleaning graphene surfaces after exposure to photoresist. Still, none of these experiments were part of the 2D-EPL MPW runs. Hence, the data should not be included in this manuscript.

In Figure 4, mobility values vary significantly depending on the wafer position, with dies at the wafer edge showing 20–30% higher mobility than those at the center. What could

be the cause of this variation? Also, how can we further improve it? 20% variation is not acceptable in industry.

We have been able to confirm that the increase in mobility closer to the flat of the wafer is related to the graphene quality and transfer. Geometry and size (200mm square) of the growth catalyst and temperature distribution and stability (heater control in the chamber) can play a role in the resulting graphene quality at the different foil areas (top vs. bottom or center vs. edge). We can only assume batch-to-batch variability due to process drift. However, the batch analyses showed that the graphene process was in spec, although with some variability within an acceptable range (acceptable for the purpose of the WPW runs, not industry production, as you correctly point out). The cleaning also led to variability in the resist residues from the transfer process, plus there was additional resist residue from the lithography processes (see also comment above about research on more efficient cleaning methods).

Automated wafer-scale transfer tools can be expected to significantly improve the quality, repeatability, and variability of the graphene across the wafers. Developing such tools with the companies Suss Microtech and EVG is a core activity in the new 2D-PL project. As explained in our answer 2 to Reviewer #1, the 2D-EPL and the 2D-PL have two main goals, the development of tools and processes (e.g., growth and transfer tools) and the offering of MPW runs (the content of our manuscript).

In general, the mobilities can be increased with optimization and professionalization of all the wafer-scale fabrication processes, including growth, transfer, encapsulation, and etching.

In addition to the die-to-die variation, a more thorough examination of wafer-to-wafer variation is required. The manuscript should provide statistically extracted performance values for at least 10 out of the 75 wafers processed during the 2D-EPL project.

We agree that an examination of wafer-to-wafer variation is desirable. Unfortunately, and we apologize for not stating this more clearly in the initial text, the 75 wafers mentioned were not all processed to the final stage. They were either used for process optimization or abandoned at some stage due to several types of failure. This number of wafers was necessary because the customer designs varied greatly from each other, making design-specific optimization difficult. Due to financial and time constraints, we stopped fabricating more wafers once a wafer with customer dies was successfully fabricated (see also our answer 2 to Reviewer #2).

We added a sentence in line 83 to clarify this:

“These wafers were not finalized, instead we used them to optimize some of the fabrication steps.”

The TLM method introduces substantial uncertainties in the extracted contact/sheet resistance values. Could the authors discuss alternative methods that may provide more reliable measurements? This discussion is crucial for offering potential customers reliable reference specifications.

We fully agree with your concern about the uncertainties of the TLM method. We have, in fact, also published several specific papers to this end^{4,5,29}. Unfortunately, TLM was the only method we had at our disposal that was applicable on a large scale. Please also

refer to our extensive discussion on the TLM method in answer 5 to Reviewer #2, and Supplementary Material 4.

A method that could be used in the future is Kelvin Probe Microscopy but we currently have no KPM tool that would work on the wafer scale and in an automatic mode. We nevertheless added this brief sentence to the manuscript at the end of the Experimental Section:

“To determine the contact resistance with more precision, Kelvin probe microscopy could be used.²⁴”.

References

1. Franklin, A. D., Han, S.-J., Bol, A. A. & Perebeinos, V. Double Contacts for Improved Performance of Graphene Transistors. *IEEE Electron Device Lett.* **33**, 17–19 (2012).
2. Schätz, J. *et al.* Button shear testing for adhesion measurements of 2D materials. *Nat Commun* **15**, 2430 (2024).
3. Sagade, A. A. *et al.* Highly air stable passivation of graphene based field effect devices. *Nanoscale* **7**, 3558–3564 (2015).
4. Venica, S. *et al.* On the Adequacy of the Transmission Line Model to Describe the Graphene–Metal Contact Resistance. *IEEE Transactions on Electron Devices* **65**, 1589–1596 (2018).
5. Driussi, F. *et al.* Dependability assessment of Transfer Length Method to extract the metal–graphene contact resistance. *IEEE Transactions on Semiconductor Manufacturing* 1–1 (2020) doi:10.1109/TSM.2020.2981199.
6. Haw-Jye Ueng, Janes, D. B. & Webb, K. J. Error analysis leading to design criteria for transmission line model characterization of ohmic contacts. *IEEE Trans. Electron Devices* **48**, 758–766 (2001).
7. Gutai, L. Statistical modeling of transmission line model test structures. I. The effect of inhomogeneities on the extracted contact parameters. *IEEE Trans. Electron Devices* **37**, 2350–2360 (1990).
8. Zhu, W. *et al.* Structure and Electronic Transport in Graphene Wrinkles. *Nano Lett.* **12**, 3431–3436 (2012).
9. Clark, K. W. *et al.* Spatially Resolved Mapping of Electrical Conductivity across Individual Domain (Grain) Boundaries in Graphene. *ACS Nano* **7**, 7956–7966 (2013).
10. Ghani, M. A. *et al.* Metal Films on Two-Dimensional Materials: van der Waals Contacts and Raman Enhancement. *ACS Appl. Mater. Interfaces* **16**, 7399–7405 (2024).
11. Chhowalla, M., Jena, D. & Zhang, H. Two-dimensional semiconductors for transistors. *Nat Rev Mater* **1**, 16052 (2016).
12. Blake, P. *et al.* Influence of metal contacts and charge inhomogeneity on transport properties of graphene near the neutrality point. *Solid State Communications* **149**, 1068–1071 (2009).
13. Xia, F., Perebeinos, V., Lin, Y., Wu, Y. & Avouris, P. The origins and limits of metal–graphene junction resistance. *Nature Nanotech* **6**, 179–184 (2011).
14. Nouchi, R., Saito, T. & Tanigaki, K. Observation of negative contact resistances in graphene field-effect transistors. *Journal of Applied Physics* **111**, (2012).
15. Zhong, H. *et al.* Realization of low contact resistance close to theoretical limit in

- graphene transistors. *Nano Research* **8**, 166–1679 (2015).
16. Chari, T., Ribeiro-Palau, R., Dean, C. R. & Shepard, K. Resistivity of Rotated Graphite–Graphene Contacts. *Nano Letters* **16**, 4477–4482 (2016).
 17. Wang, W., Muruganathan, M., Kulothungan, J. & Mizuta, H. Study of dynamic contacts for graphene nano-electromechanical switches. *Japanese Journal of Applied Physics* **56**, 04CK05 (2017).
 18. Thissen, N. F. W. *et al.* Graphene devices with bottom-up contacts by area-selective atomic layer deposition. *2D Materials* **4**, 025046 (2017).
 19. Cusati, T. *et al.* Electrical properties of graphene -metal contacts. *Scientific Reports* **7**, 5109.
 20. Gahoi, A. *et al.* Contact resistance study of various metal electrodes with CVD graphene. *Solid-State Electronics* **125**, 234–239 (2016).
 21. Gahoi, A. *et al.* Dependable Contact Related Parameter Extraction in Graphene–Metal Junctions. *Adv. Electron. Mater.* **6**, 2000386 (2020).
 22. Driussi, F. *et al.* Dependability Assessment of Transfer Length Method to Extract the Metal–Graphene Contact Resistance. *IEEE Transactions on Semiconductor Manufacturing* **33**, 210–215 (2020).
 23. Venica, S. *et al.* On the Adequacy of the Transmission Line Model to Describe the Graphene–Metal Contact Resistance. *IEEE Transactions on Electron Devices* **65**, 1589–1596 (2018).
 24. Shaygan, M. *et al.* Low Resistive Edge Contacts to CVD-Grown Graphene Using a CMOS Compatible Metal. *Annalen der Physik* **529**, 1600410 (2017).
 25. Illarionov, Yu. Yu. *et al.* Bias-temperature instability in single-layer graphene field-effect transistors. *Applied Physics Letters* **105**, 143507 (2014).
 26. Alexander-Webber, J. A. *et al.* Encapsulation of graphene transistors and vertical device integration by interface engineering with atomic layer deposited oxide. *2D Mater.* **4**, 011008 (2016).
 27. Hong, J. *et al.* Origin of New Broad Raman D and G Peaks in Annealed Graphene. *Sci Rep* **3**, 2700 (2013).
 28. Gong, C. *et al.* Rapid Selective Etching of PMMA Residues from Transferred Graphene by Carbon Dioxide. *J. Phys. Chem. C* **117**, 23000–23008 (2013).
 29. Cheng, Z. *et al.* How to report and benchmark emerging field-effect transistors. *Nature Electronics* **5**, 416–423 (2022).